# Engineering 3D-Printed Bioresorbable Scaffold to Improve Non-Vascularized Fat Grafting: A Proof-of-Concept Study

**DOI:** 10.3390/biomedicines11123337

**Published:** 2023-12-18

**Authors:** Amélia Jordao, Damien Cléret, Mélanie Dhayer, Mégann Le Rest, Shengheng Cao, Alexandre Rech, Nathalie Azaroual, Anne-Sophie Drucbert, Patrice Maboudou, Salim Dekiouk, Nicolas Germain, Julien Payen, Pierre Guerreschi, Philippe Marchetti

**Affiliations:** 1UMR9020–UMR-S 1277–Canther–Cancer Heterogeneity, Plasticity and Resistance to Therapies, CNRS, Inserm, CHU Lille, Oncolille, University Lille, F-59000 Lille, France; amelia.jordao@lattice-medical.com (A.J.); nicolas.germain@inserm.fr (N.G.); 2Lattice Medical, 80 rue du Docteur Yersin, F-59120 Loos, France; 3University of Lille, Faculté de Pharmacie, Plateau RMN, UFR3S, F-59000 Lille, France; 4University of Lille, ULR 7365–GRITA–Groupe de Recherche Sur Les Formes Injectables Et Les Technologies Associées, F-59000 Lille, France; nathalie.azaroual@univ-lille.fr; 5U 1008 Controlled Drug Delivery Systems and Biomaterials, Inserm, F-59000 Lille, France; 6Service de Biochimie, CHU Lille, F-59000 Lille, France; patrice.maboudou@chu-lille.fr; 7Centre de Bio-Pathologie, Banque de Tissus, CHU Lille, F-59000 Lille, France; 8Service de Chirurgie Plastique, CHU Lille, F-59000 Lille, France

**Keywords:** autologous fat grafting, tissue engineering, 3D printing, chorioallantoic membrane, vascularization, biomaterials

## Abstract

Autologous fat grafting is the gold standard for treatment in patients with soft-tissue defects. However, the technique has a major limitation of unpredictable fat resorption due to insufficient blood supply in the initial phase after transplantation. To overcome this problem, we investigated the capability of a medical-grade poly L-lactide-co-poly ε-caprolactone (PLCL) scaffold to support adipose tissue and vascular regeneration. Deploying FDM 3D-printing, we produced a bioresorbable porous scaffold with interconnected pore networks to facilitate nutrient and oxygen diffusion. The compressive modulus of printed scaffold mimicked the mechanical properties of native adipose tissue. In vitro assays demonstrated that PLCL scaffolds or their degradation products supported differentiation of preadipocytes into viable mature adipocytes under appropriate induction. Interestingly, the chorioallantoic membrane assay revealed vascular invasion inside the porous scaffold, which represented a guiding structure for ingrowing blood vessels. Then, lipoaspirate-seeded scaffolds were transplanted subcutaneously into the dorsal region of immunocompetent rats (*n* = 16) for 1 or 2 months. The volume of adipose tissue was maintained inside the scaffold over time. Histomorphometric evaluation discovered small- and normal-sized perilipin^+^ adipocytes (no hypertrophy) classically organized into lobular structures inside the scaffold. Adipose tissue was surrounded by discrete layers of fibrous connective tissue associated with CD68^+^ macrophage patches around the scaffold filaments. Adipocyte viability, assessed via TUNEL staining, was sustained by the presence of a high number of CD31-positive vessels inside the scaffold, confirming the CAM results. Overall, our study provides proof that 3D-printed PLCL scaffolds can be used to improve fat graft volume preservation and vascularization, paving the way for new therapeutic options for soft-tissue defects.

## 1. Introduction

Today, autologous fat grafting is a widespread practice in the field of reconstructive and aesthetic surgery [1]. Fat transfer involves harvesting autologous non-vascularized fat from a donor site, such as the abdomen or trochanteric regions, and injecting it into another site that needs filling. Fat grafting is widely used to repair defects after tumor removal in post-cancer breast reconstruction, to address post-traumatic defects, and in the treatment of congenital anomalies [1]. This is also a valuable method for reconstructing subcutaneous layers in severe burn scars [2], which is particularly important for maintaining the mobility and the quality of the skin. Fat grafting possesses the advantages of being a natural, biocompatible technique, as well as a minimally invasive and easily accessible surgery. Further, grafting is thought to be less painful than vascularized flap surgery [3]. Although fat grafting remains the gold standard for soft-tissue reconstruction, this method results in highly unpredictable long-term results due to high rates (30–70%) of volume resorption in the grafted site [4]. Due to the erratic nature of fat survival, iterative lipofilling sessions may be required to achieve the desired volume. This can increase the overall cost, surgical risks, and discomfort for the patients [1].

Several clinical and biological factors contribute to excessive fat graft resorption. The physical handling of fat tissue during harvesting, processing, and injection largely impacts the viability of transferred adipocytes. The mechanical disruption, injection technique, and the pressure applied during fat transfer cause irreversible damages to delicate mature adipocytes, reducing their survival rate [5]. Moreover, adipose tissue trauma triggers an acute inflammatory response that contributes to the resorption of grafted fat via the activation of immune cells and production of inflammatory mediators that attack transferred fat. In this way, grafted adipocytes and the inflammatory process create a vicious cycle that eventually augments the resorption rate of grafted fat [6]. Additionally, the transferred adipocytes initially lack direct blood supply. Studies have shown that, under favorable conditions, neovascularization begins only 2 or 3 days post-injection [6]. During the early stages, the survival of grafted fat depends on obtaining oxygen and nutrients via diffusion from surrounding tissue. However, this diffusion process is limited and the lack of robust blood supply immediately after injection leads to local hypoxia, causing extensive cell death. Adipose tissue grafts survive for less than 3 days in ischemic conditions [7]. Finally, the altered environment and nutrient availability within the recipient site can affect the metabolic activity of adipocytes. This, in turn, may contribute to their resorption. It has been observed that grafted adipocytes increase lipolysis alongside a concomitant decrease in lipogenesis, metabolic changes which may accelerate fat resorption [8].

Several approaches have been developed to improve fat graft survival, including changes to procurement and processing techniques [9,10]. Alternatively, patients can resort to tissue engineering techniques with the potential to reduce fat resorption in improving the long-term maintenance of transferred adipocytes [2,11,12]. It is possible to imagine designing scaffold materials that provide structural and mechanical support to create a three-dimensional framework that mimics the natural environment of fat tissue. Such scaffolds create spaces for transferred fat cells, promoting their integration within the surrounding tissues, facilitating the establishment of new blood vessels, and eventually maintaining the volume and shape of the injected adipose tissue [13]. In order to meet the requirements for clinical use, scaffolds for soft-tissue repair should meet the following minimum specifications: possessing mechanical properties similar to those of native adipose tissue; proper biocompatibility; and bioresorbability with non-toxic degradation products, serving as the long-term support of autologous tissue volume without risk of chronic inflammation [14]. The advent of scaffold technology in fat grafting represents a groundbreaking shift in the realm of reconstructive and cosmetic surgery. This innovative approach addresses the longstanding challenges associated with traditional fat grafting methods, such as unpredictable survival rates of the grafted fat and inconsistent results.

Both synthetic and natural polymers can be used to fabricate scaffolds for soft-tissue repair [15]. In the field of adipose tissue regeneration, injectable hydrogels are emerging as a promising tool due to their ability to mimic the extracellular matrix, supporting cell survival and enhancing the effectiveness of autologous fat transfer. Research has shown that hydrogels composed of materials like collagen, hyaluronic acid, matrigel, and fibrin can significantly improve the survival rate of fat grafts. Hydrogels can be engineered to have specific mechanical properties and to release bioactive molecules, aiding in tissue regeneration [16]. In a notable study, Yuan et al. developed a biomimetic peptide nanogel that exhibits antimicrobial properties and aids in hemostasis, promoting the healing of chronic epidermal wounds [17]. Bioresorbable synthetic polymers offer several advantages in such repair efforts compared to natural materials. These polymers can be synthesized with precise control over their composition, structure, and properties, ensuring reproducibility and reliability of the materials for soft-tissue repair applications [18]. This also allows for the tailoring of material properties such as mechanical strength, flexibility, or degradation rate to match the specific requirements of scaffolds. It is noteworthy that synthetic polymers can be used as ink materials for 3D printing to create complex internal structures with controlled pore sizes within soft-tissue repair scaffolds [19,20]. Among them, PLCL (poly (L-lactide-co-ε-caprolactone)) copolymers are synthetic polymers that have gained our attention for their potential use in soft-tissue repair scaffolds [21,22]. These bioresorbable polymers possess relevant stiffness properties, excellent biocompatibility and a history of safe use in various medical applications. Additionally, they are compatible material for fuse deposition modeling and 3D printing [23]. They also offer a compromise between the stiffness required to maintain adipose tissue and the flexibility needed for this type of cell, making them suitable for soft-tissue repair applications that require moderate mechanical properties [23].

Therefore, the aim of this study was to present in vitro and in vivo proof of concept for the use of 3D-printed PLCL bioresorbable scaffolds to improve the efficacy of fat grafting.

## 2. Materials and Methods

### 2.1. 3D Printing of PLCL Scaffolds

Bioresorbable medical-grade PLCL (poly-L-lactide-co-caprolactone, 70:30 ratio) was obtained from EVONIK (Darmstadt, Germany) or PURAC (CORBION, Gorinchem, The Netherlands. PLCL pellets were stored at −20 °C and were dried at 40 °C during 24 h pre-processing. PLCL was first melt-spun into 2.85 mm filaments using a Composer Series 350 extruder (3DEVO, Utrecht, The Netherlands). Scaffolds were designed using Solidworks Software (Version 2019 SP4; Waltham, MA, USA); these were then printed using a Ultimaker S5 FDM (Utrecht, The Netherlands) (see Table 1 for design characteristics). Scaffolds were sterilized with ethylene oxide (STERISERVICES, Bernay, France). Aeration was then performed for 72 h to remove ethylene oxide residues. Scaffolds were stored at −20 °C inside a blister until experiments were performed.

### 2.2. Mechanical Characterization and Analysis of Microarchitecture via High-Resolution X-ray Tomography

For mechanical characterization, compression specimens, fabricated following the ASTMD1621-16 standard, were placed in a tensile bench equipped with a force transducer (Interface, Atlanta, GA, USA) and a compression plate. A monotonic, displacement-controlled compression test was performed at a constant rate of 2.6 mm/min with a 100 N cell. Compression specimens were initially weighed and soaked in PBS, before being kept at 37 °C. At indicated time points (from 1 h to 72 h), taken out; excess liquid was wiped off using a paper towel and specimens were weighed. PBS uptake (%) = $\frac{Mt-M0}{M0}\times 100$, where *Mt* is the weight at time *t* and *M*0 is the weight of dry specimens. Alternatively, volumes generated via high-resolution X-ray tomography of microarchitectural structures of 3D scaffolding were obtained using a UltraTom phase contrast (Rx-solutions, Chavanod, France). Data underwent computed tomography at the ISIS4D X-ray CT platform Univ Lille. A 3D scaffold was placed perpendicularly to the X-ray beam, the spatial resolution was set to 33 μm, and the acquisition time was approximately 25 min per sample.

### 2.3. In Vitro Scaffold Degradation

Sterile PLCL scaffolds were degraded in a DMEM culture medium (Gibco, Thermofisher, Waltham, MA, USA) without penicillin–streptomycin (PS) (Gibco, Thermofisher, Illkirch-Graffenstaden, France) or FBS (Eurobio Scientific, Les Ulis, France), pH 7.4 ± 0.2, at 37 °C under agitation at a constant speed (70 rpm) for the indicated times. An extraction ratio of 1:3 (DMEM volume (mL): area (cm^2^)) was used according to the national standards ISO10993-12:2012. The pH was adjusted to pH 7.4 ± 0.2. Media extracts were used for the identification of degradation products and for cell culture. Scaffolds were recovered in order to evaluate macroscopic changes.

### 2.4. Identification and Quantification of Degradation Products via Quantitative Proton Nuclear Magnetic Resonance (^1^H-NMR)

Media extracts of 500 µL were transferred to an NMR tube and supplemented with 50 µL of deuterated water (D2O). Filtered (0.22 µm filters) and unfiltered extraction media were analyzed. All experiments were performed at 22 °C using a BRUKER NEO 500 MHz NMR spectrometer (Bruker, MA, USA), with a 5 mm TXI probe. Spectra were acquired with suppression of water signal. The pre-saturation of the water signal was achieved with a zgpr sequence before the pulse. The 90° pulse length was calibrated and adjusted based on each sample. The number of transients was 32 and a repetition time of 15 s was used. The data were processed using software TOPSPIN 4.1.4 (Bruker, MA, USA). A line broading (0.3 Hz) apodization was applied to all FIDs before Fourier transformation. Experiments were acquired under quantitative conditions, and Module ERETIC2 was used to calculate the concentration of each sample. Molecules were identified with two-dimensional 1H/13C correlation spectroscopy (HSQC and HMBC) and selective TOCSY experiments on the isolated signals to observe scalar couplings in small molecules and establish 1H-1H connectivity.

### 2.5. Cell Culture

We cultured 3T3-L1 preadipocytes (ATCC-CL-173 from ATTC, Manassas, VA, USA) in DMEM High Glucose medium (Gibco, Thermofisher, Illkirch-Graffenstaden, France) with 10% FBS (Eurobio scientific, Les Ulis, France) and 1% PS (Gibco, Thermofisher, Illkirch-Graffenstaden, France) at 37 °C, 5% CO_2_. After reaching 80% confluence, cells were rinsed using PBS (Thermofisher) and detached with 0.25% trypsin-EDTA (Thermofisher). Pre-adipocytes were seeded at a density of 3 × 10^3^ cells/cm^2^ in a 24-well plate (1 mL/well). Differentiation was induced in confluent cells (at D0 3 days after seeding) by replacing DMEM with differentiation medium. Differentiation of 3T3-L1 was initiated in DMEM complete medium using an adipocyte differentiation cocktail containing 1 µM Dexamethasone, 0.5 mM 3-Isobutyl-1-methylxanthine, and 10 µg/mL human insulin, all sourced from Sigma-Aldrich, St. Louis, MO, USA. After 4 days (D4), cells were switched to DMEM complete medium supplemented with 10 µg/mL insulin for an additional 3 days. Throughout this 7-day period of differentiation, 1 mL of media extracts from degraded scaffolds was added at D0 and D4 of differentiation. At D7 of differentiation, cell supernatants were recovered for analysis. Additionally, cells were prepared for the quantification of differentiation and adipocyte viability.

### 2.6. Assessment of Viability and Adipocyte Differentiation

At D7 of differentiation, cells were labeled with Hoechst 33342 (1 µg/mL, Thermofisher), BODIPY 493/503 ^TM^ (1 µg/mL, Thermofisher), and/or propidium iodide (1 µg/mL, Sigma-Aldrich) for 20 min at 37 °C in the dark. After labelling, the cells were rinsed with PBS and label-free DMEM was added before immediate analysis. Adipocytes were visualized and quantified via fluorescence microscopy using the BIOTEK^TM^ Cytation 5 multimodal plate reader (Agilent, CA, USA). For LDH measurement, 400 µL of supernatants was used to determine LDH activity via UV photometric assay (Cobas 8000 modular analyzer, Roche Diagnostics, Mannheim, Germany).

### 2.7. Cell Seeding Experiment

Differentiation on the 3D-printed PLCL scaffolds was assessed by culturing 3T3-L1 preadipocytes on scaffolds in DMEM with differentiation medium for 7 days. Briefly, sterile scaffolds were pre-wetted for 24 h in DMEM medium prior to cell seeding. Subsequently, 3T3-L1 preadipocytes were seeded onto scaffolds in 6-well plates at a cell density of 1 × 10^6^ cells/scaffold/well; then, they were differentiated as described above. At D7, cells on scaffolds were labelled with fluorescent dyes and visualized using a Biotek Cytation 5^TM^. Alternatively, cell seeding efficacy was assessed 3 h after the incubation of 3T3-L1 cells on scaffolds (1 × 10^6^ cells/scaffold/well), as described previously [24]. The cell seeding efficacy was calculated using the following formula: cell seeding efficacy (%) = $\frac{N0-Nt}{N0}\times 100$, where *N0* is the total number of initial cells and *Nt* is the number of cells not attached to the scaffolds, instead remaining at the bottom of the well.

### 2.8. CAM Assay

Sterile pre-degraded and undegraded scaffolds (1.5 × 1.5 cm) were implanted into chicken chorioallantoic membranes (CAM). Fertilized eggs were received and incubated in an automatic incubator (JANOEL24, Wiltec Wildanger Technik GmbH, Eschweiler, Germany) for 3 days in conditions of 37 °C and 60% humidity. They were then opened. Scaffolds, rinsed with PBS, were implanted at day 10 of embryonic development into the CAM for 4 days. Images were taken using a Motic binocular loupe SMZ-171 (Motic, Wetzlar, Germany) equipped with a Moticam 1080. Vascular density and vessel diameter were quantified using ImageJ 1.53t software (National Institutes of Health, Bethesda, MD, USA).

### 2.9. Animal Models

A total of 16 female *Wistar* rats aged 12 weeks, with an initial weight of 250 g, were used. Rats were randomly divided into 2 subgroups of 8 each. These groups differed by the duration of the implantation, i.e., 1 or 2 months. Rats were implanted into the dorsal region subcutaneously, bilaterally, with a 6 mm thickness/8 mm diameter scaffold on the left side and a 2 mm thickness/8 mm diameter scaffold on the right side. Before implantation, each scaffold was filled in vitro with fat harvested from adipose tissue previously collected from the animal’s posterior flanks. Fat was fractionated and centrifuged to eliminate debris and unwanted fibrotic tissue, producing lipoaspirate. In addition, the same volume of lipoaspirate alone was implanted on each side of the animal and served as the control. All implantations were performed under sterile conditions and after inhalation of volatile anesthetic agent (5% isoflurane for induction, followed by 2–3% during the entire surgical intervention).

### 2.10. Tissue Assessment for Histomorphometry and Immunohistochemistry

At the time of euthanasia, the macroscopic evaluation of fat tissues was performed. Then, tissues were harvested and immediately fixed with 10% paraformaldehyde overnight before paraffin embedding. Histological specimens were routinely stained with hematoxylin and eosin or Masson’s trichrome, as previously detailed [25,26]. Histological tissues were also stained with anti-perilipin antibody, TUNEL staining, anti-CD31 antibody, or anti-CD68 antibody, as previously described [25,26].

### 2.11. Statistical Analysis

Graphpad Prism 8^®^ software (Graphpad Software^TM^, Inc., San Diego, CA, USA) was used for the statistical study of all results. Mechanical (compressive modulus and water uptake) factors, adipocyte differentiation, and viability results are expressed as mean ± SD. Results of cell number for cell seeding efficacy are expressed as whisker bar graphs (median, 5–95 percentile—max–min), as are angiogenesis on CAM model and histomorphometric and immunohistochemistry results (median, 5–95 percentile max–min). All data were analyzed via non-parametric one-way analysis of variance (Kruskal–Wallis test), followed by a Dunn’s test, in order to compare groups at all time points, and non-parametric *t* tests (Mann–Whitney test), allowing us to compare two groups at a single time point. The level of significance was set at * *p* ≤ 0.05.

## 3. Results

### 3.1. Design of the Scaffolds and Mechanical Characterization

Scaffold design has been demonstrated to have a major impact on adipose tissue engineering [27]. Scaffolds were 3D printed with a high degree of accuracy (see Video S1). FDM 3D printing was carried out using 0.2 mm PLCL filaments, contributing to designing a homogeneously organized network with interconnected pores (see Video S1). As shown in the macroscopic and microscopic images (Figure 1A–D), we designed porous structures by interlacing regular filament loops, a structure which is supposed to be favorable for fat grafting and the development of neovascular networks [27]. It is noteworthy that the PLCL filament thicknesses across the scaffolds were homogeneous, as shown in Figure 1D. We opted for the design of a scaffold with a large pore size (Figure 1E) and overall porosity > 90% (Table 1), a prerequisite for cell migration and infiltration into the 3D scaffold. Thus, we fabricated a flexible multilayer structure (Figure 2A) with a density lower than 0.1 g/cm^3^. The results of the compression tests (Figure 2B) performed at room temperature in dry conditions show that the initial compressive modulus (E_i_) of the scaffold was 88 kPa ± 12 (Figure 2C). After immersion in PBS at 37 °C (wet condition), simulating body conditions, the scaffold was much softer, with an E_i_ decreased by 89%, reaching values closer to those of adipose tissue (1–20 kPa) [28]. The PLCL scaffold was characterized by progressive PBS uptake (Figure 2D) and good mechanical properties, making it suitable for use as a scaffold in adipose tissue regeneration.

### 3.2. In Vitro Degradation Profile of PLCL Scaffolds

The degradation of PLCL scaffolds was carried out in vitro in DMEM medium (Figure 3A). Then, scaffolds were analyzed at the indicated times (Figure 3B) and ^1^H-NMR analysis of the supernatant was used to determine the composition of degradation products over time (Figure 3C). PLCL scaffolds showed progressive degradation during the 24-week in vitro study, as seen in Figure 3B. Macroscopically, scaffolds were seen to retain their general shape and structure until week 4. This was followed by progressive erosion beginning at the edges, with no voids found towards the core of the printed fibers. From week 20, scaffolds became considerably thinner and translucent in central areas, which made them more susceptible to crumbling, causing non-uniform structures (Figure 3B). Another indication of scaffold degradation was the ^1^H-NMR profile of the media extracts (Figure 3C). These spectra confirm that the PLCL scaffolds degraded progressively, as expected [29]. Based on the characteristic spectra peaks of PLA and PCL and 6-hydroxyhexanoic acid components, lactate was found in the degradation media at a time point as early as week 8. Conversely, PCL + 6-hydroxyhexanoic acid appeared later at week 16, as indicated by specific chemical shifts (Table 2), suggesting that LA moieties degraded faster than CL components in PLCL scaffold due to their hydrophilicity [29]. After 24 weeks of degradation, the concentrations of lactate and PCL + 6-hydroxyhexanoic acid were 11.3 mM and 3.6 mM, respectively.

### 3.3. In Vitro Cellular Effects of PLCL Scaffold Degradation Products

To determine the biocompatibility of the PLCL scaffold, media extracts of PLCL scaffolds, harvested at different time points of degradation (Figure 3A), were incubated in the presence of 3T3-L1 pre-adipocytes undergoing adipocyte differentiation (Figure 4A). Then, cells were analyzed for differentiation (Figure 4B,C) and to assess viability (Figure 4D–F). With the exception of week 24, the exposure of PLCL extracts harvested at all timepoints did not significantly affect adipocyte differentiation compared with the control group, as judged by the presence of lipid accumulation in differentiated adipocytes after BODIPY 493/503 staining (Figure 4B,C). Indeed, only media extracts from the most degraded scaffolds (for 24 weeks) significantly reduced the production of mature adipocytes (Figure 4B,C). Cytotoxicity evaluation via PI staining indicated that adipocyte viability, when exposed to the extract of degrading PLCL scaffolds, exceeded 90%. This was similar to the viability of adipocytes cultured using blank culture medium (Figure 4D,E). These results were confirmed when cytotoxicity was assessed using lactate dehydrogenase (LDH) released from adipocytes into the culture media (Figure 4F), attesting the absence of the toxicity of extracts of degrading PLCL scaffolds.

### 3.4. Cell Attachment on PLCL Scaffold

Next, cell behavior after cell–PLCL fiber interaction was studied by culturing undifferentiated or differentiated 3T3-L1 cells on PLCL scaffold, which was in accordance with the specific protocol shown in Figure 5A. We assessed the initial cell attachment after 3 h of cell seeding by counting the unattached cells. We found a cell seeding efficacy of 27.6% ± 3%, consistent with the complex design of a scaffold with interconnected pores (Figure 1). Microscopic images confirmed the attachment of undifferentiated and differentiated 3T3-L1 viable cells along the PLCL fibers, as judged by the attachment of BODIPY-positive PI-negative cells (Figure 5B). Therefore, our results suggest that 3T3L1 are able to differentiate into viable mature adipocytes on a PLCL scaffold.

### 3.5. Vascularization of the PLCL Scaffolds in the CAM Assay

In vivo, the success of fat engraftment critically depends on the revascularization process through neovascularization, which should start within 7 days of grafting [7]. Thus, we decided to study the vascular colonization of the 3D PLCL scaffolds, which were undegraded and pre-degraded at early time points (week 4 and 8) when placed in a highly vascularized environment using the ex ovo chick chorioallantoic membrane (CAM) assay (Figure 6A). As observed in Figure 6B, macroscopical evaluation clearly showed that PLCL scaffolds did not elicit any adverse reaction and did not alter blood vessels in the CAM, confirming the biocompatibility of the material used (Figure 6B). Above all, the CAM assay shows intense infiltration of blood vessels within undegraded or pre-degraded scaffolds. In a situation of PLCL filament interlacing, blood vessels penetrated to the center of scaffolds, and vessel infiltration was present throughout the depth of scaffolds (Figure 6B). Vascular infiltration was identical regardless of the scaffolds used, with comparable vascular densities within the different scaffolds (Figure 6C). Interestingly, there was a directed growth of capillary vessels, which seemed to follow the axis of the PLCL filaments (Figure 6B yellow arrows). Thus, these data indicate that PLCL scaffolds are fully biocompatible and should provide favorable volume for initial neovascularization of grafted tissue, allowing them to be considered for testing on in vivo animal models.

### 3.6. In Vivo Implantation of Lipoaspirate-Seeded Scaffolds

Next, we assessed in vivo whether the lipoaspirate-seeded PLCL scaffold could define and maintain the adipose tissue volume desired. To do that, lipoaspirate-seeded PLCL scaffolds (2 mm or 6 mm thickness) were implanted into the dorsa of rats for 1 or 2 months and the results were compared to those obtained via fat injection alone, the technique used as a control (Figure 7A). The macroscopic inspection of explanted tissues did not a reveal significant pattern of inflammation or the presence of necrotic tissues (Figure 7B). As shown in Figure 7C, HE staining revealed normal adipose tissue organization with lobular architecture within the PLCL scaffolds after 1 or 2 months. Importantly, the volume of adipose tissue was maintained inside the scaffold for up to 2 months after implantation (Figure 7C). No adipocyte was observed as migrating out of the scaffolds into the surrounding tissues. Conversely, in the control condition without scaffold, the residual volume of adipose tissue was thin and flattened (resorption rate close to 100%) as early as 1 month after implantation (Figure 7C). Adipose cell number per mm2 of tissue within the scaffolds remained constant over time (Figure 7D). Indeed, limited cell death was detected via terminal deoxynucleotidyl transferase dUTP nick-end labeling (TUNEL) staining (Appendix A). The adipose tissue was composed mainly of adipocytes of smaller size (no hypertrophy) (Figure 7E). Most of these expressed *perilipin*, a marker of cell maturity (Figure 7F). Mature viable adipocytes were present in the center region of the scaffolds, as well as at the edges (Figure 7C,F). Masson’s trichrome staining method was used to evaluate the local–foreign body reactions of the PLCL scaffold. (Figure 8A). In the peripheral zone, a thin encapsulation of the adipose tissue was found to possess a fibrous connective tissue rim (Figure 8A, arrows). We also identified this from Masson’s trichrome staining layers of collagen fibers (blue staining), which is associated with the infiltration of CD68^+^ macrophages (Figure 8A) in the form of specific spots around the PLCL fibers. However, the surface area of connective tissue remained small and shrank over time (Figure 8B). Finally, CD31 immunostaining clearly showed the presence of well-developed blood vessels inside the adipocyte-seeded PLCL scaffolds (Figure 9A), corroborating the CAM findings (Figure 6). The presence of a high density of vessels within the adipose tissue persisted over time (Figure 9B), and their location was found in all adipose zones within the scaffold (Figure 9C). Overall, these results indicate that a PLCL scaffold allows practitioners to retain the predefined shape and dimensions of viable mature adipose tissue after in vivo implantation.

## 4. Discussion

Fat survival and regeneration are limiting factors in fat grafting [1]. Previous studies have demonstrated a growing interest in adipose tissue engineering in order to solve the above problem [12]. The use of fat-seeded scaffolds, providing both sufficient space for blood vessel ingrowth and mechanical support to adipose tissue, could constitute a solution to maintain long-term adipose tissue survival [27]. In this paper, we provided proof of concept for the use of PLCL scaffolds to maintain the shape and volume of viable adipose tissue over time. The use of PLCL scaffolds fulfils several goals: (i) mimicking the mechanical properties of native adipose tissue, (ii) supporting in vivo neovascularization, and (iii) supporting cell infiltration and differentiation in viable mature adipocyte.

First, scaffold mechanical properties, which are strictly dependent on the degree of scaffold porosity, 3D design, and the nature of polymers used, exert influence on cell fate and adipose tissue preservation [24,30]. Here, we demonstrated that PLCL scaffolds have an appropriate design, with mechanical features close to those of adipose tissue. A scaffold matching the mechanical characteristics of adipose tissue, i.e., having a compression modulus in the range of 1–20 kPa [28], can stimulate differentiation toward the adipocyte lineage by mimicking the stiffness of the native tissue [31]. Conversely, excessive mechanical stresses inhibit adipogenesis [32]. The compressive modulus of the PLCL scaffold used in this study was slightly higher than that of adipose tissue. Nevertheless, one could expect, with the progressive degradation of PLCL fibers over time, that the compressive modulus decreased to the value of the grafted tissue.

Second, we decided to fabricate PLCL scaffolds with highly porous architectures in order to allow adequate neovascularization in the fat graft and thus prevent its accelerated resorption. Grafted fat initially lacks vascular support, and its survival is critically dependent on successful revascularization [7]. The simple diffusion of oxygen and nutrients from surrounding tissue is generally insufficient to feed large volumes of adipose tissue as the maximum diffusion distance for oxygen has been estimated to be around 150 µm [33]. Low oxygen pressures and/or the absence of nutrients in the center of the graft can lead to extensive adipocyte necrosis. It is well established that scaffold morphology has a crucial impact on adipose tissue viability. Indeed, scaffolds with small pores do not permit the penetration of vessels and are unsuitable for the survival of large volumes of adipose tissue. The minimum pore size favorable for adipose tissue engineering [34] is 120 µm. Therefore, we produced scaffolds containing large, interconnected pores to allow early vascular ingrowth into the scaffold. Interestingly, during the CAM assay and after in vivo implantation of PLCL scaffolds, we observed extensive blood vessel infiltration, suggesting that a proper nutrient and oxygen delivery is possible even in the inner core of the scaffold.

Third, several tissue engineering strategies have been developed to enhance fat graft survival and therefore allow for the maintenance of tissue volume. These include the use of natural biomaterials [35] and/or growth factors [36]. Although very attractive, these approaches are often difficult to integrate into clinical practice due to regulatory constraints and potential risk. In this study, we demonstrated that a PLCL scaffold is able to maintain an adipose tissue volume over time and could be an alternative solution to the above-mentioned strategies. Indeed, PLCL scaffold supported the differentiation of preadipocytes in vitro. When implanted in vivo, scaffolds favored adipocyte survival and organization in adipose tissue. This was even true in the inner core of the scaffold associated with intense vascular supply. In addition, we observed a large number of small-sized adipocytes. These could represent progenitor cells, able to regenerate adipose tissue during tissue remodeling [25].

The selection of synthetic polymers to fabricate the scaffold in adipose tissue regeneration involves the careful consideration of various criteria to ensure optimal performance and biocompatibility. Bioresorbable PLGA scaffolds have been applied with success for the long-term maintenance of adipose formations [37]. The copolymer PLCL has been used as a tissue engineering scaffold for bone, ligament, orthopedic, skin, nerve, and vascular tissue engineering [38]. Poly(D,L-lactide) (PDLA) and poly(ε-caprolactone) (PCL) were used to print 3D scaffolds responsible for in vivo breast tissue regeneration [27,39]. Furthermore, poly (Llactide-co-trimethylene carbonate) (PLATMC) and poly(L-lactide-co ε-caprolactone) (PLCL) were also suitable for soft-tissue regeneration [30]. Our study presented the copolymer PLCL as a promising candidate in fabricating scaffold for adipose tissue engineering for the following reasons: *(i) biocompatibility*. Our results demonstrated that medical grade PLCL did not elicit adverse severe immune response or toxic effects that could hinder tissue regeneration. However, PLCL had no negative effect on adipocyte differentiation in vitro after 20 weeks of degradation, although a slight but significant decrease in the rate of adipocyte differentiation was observed after 24 weeks of degradation. This decrease was not accompanied by a loss of viability and might be linked to the loss of medium stability and not only to the presence of degradation products. It is well-established that the degradation products of the polymers PLA and PCL is not harmful to the body since they are easily metabolized or excreted [40]. Nevertheless, we observed a mild inflammation (presence of CD68 macrophages) with intermediate fibrous tissue formation. These effects were transient and significantly reduced in the time point at 2 months, indicating long-term safety. Interestingly, it has previously been established that mild inflammatory process can benefit adipose tissue maintenance since inflammation is involved in blood vessel formation [41]. Thus, our histological findings suggest that the ingrowth of blood vessels inside the connective tissue around the scaffold preceded vascular budding within it. The second factor motivating our characterization of copolymer PLCL was *(ii) degradation rate*. For optimal results, the polymer degradation rate should match the rate of adipose tissue regeneration. Too rapid polymer degradation leads to premature loss of scaffold mechanical properties, while excessive polymer persistence causes strong inflammatory effects that are detrimental to adipose tissue growth. Six months would represent the ideal degradable time for fat tissue regeneration [42]. Our results indicate that PLCL scaffolds maintained their structure after 2 months in vivo and at least in part after 6 months in vitro, which was favorable for the regeneration of adipose tissue without deleterious effects. The third factor involved was the *(iii) manufacturing process*. The co-polymer PLCL has the major advantage of being suitable for use in a variety of manufacturing techniques such as electrospinning or FDM 3D printing to produce customized scaffolds [43,44]. In this study, we demonstrated that 0.2 mm thick PLCL filaments can be successfully printed in porous scaffolds with different interconnected pore sizes. PLCL scaffolds were prepared via fused deposition modelling (FDM), allowing us to produce 3D implantable scaffolds to exactly match patient defects in a reproducible manner with a very attractive cost–effectiveness ratio. The final factor considered was *(iv) regulatory approval.* PLA and PCL are used in orthopedic and disc implants and are approved by the US Food and Drug Administration for clinical applications [45]. Thus, we developed a PLCL scaffold that is easy to handle in the clinic and able to regenerate lost adipose tissue volumes in a simple, effective, and cost-effective autologous manner.

However, our study had some limitations. There was a limitation in relation to the size of scaffold used. The volume of the designed scaffolds (100 and 300 mm^3^) was scaled down for the rodent experiment. Research is currently being conducted on a porcine model with scaffold volumes close to those used in humans, granting the advantage of being able to be implanted. The study limitations also included the follow-up duration (2 months), which was insufficient for studying the complete degradation of the PLCL scaffold and therefore for observing potential late complications, if any, such as the loss of cell viability or fibrosis. Therefore, more studies are needed to verify the effectiveness and safety of PLCL scaffolds.

Irrespective of these considerations, the presented study provides proof of concept that a 3D-printed PLCL scaffold, by providing a structured and supportive environment, can be used for the efficient regeneration of vascularized mature viable adipose tissue, paving the way for future human clinical trials. Scaffolds offer an engineered matrix that supports the structural integrity of the grafted by facilitating vascularization. This advancement should ensure a more reliable and uniform integration of the graft with the recipient tissue, leading to superior aesthetic and functional results.

## Figures and Tables

**Figure 1 biomedicines-11-03337-f001:**
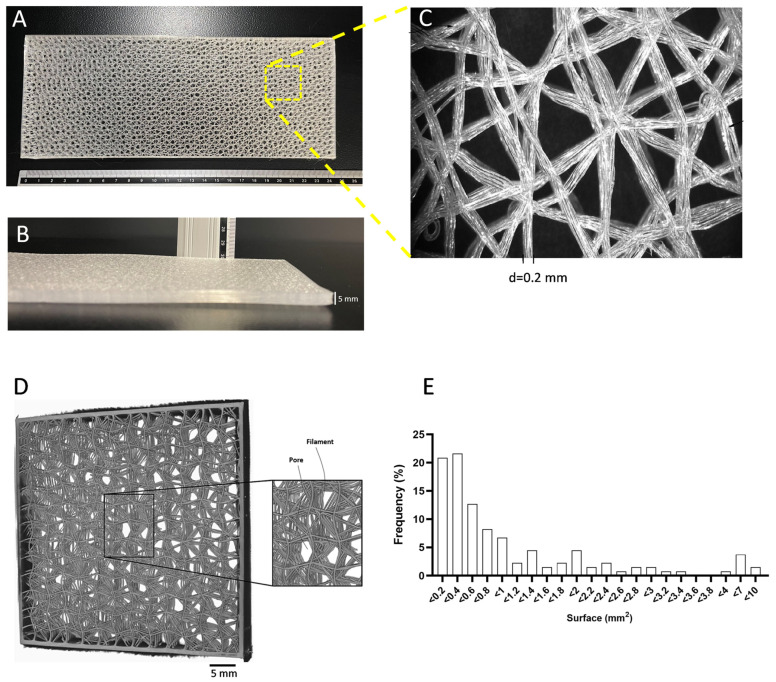
3D-printed PLCL scaffold design. Macroscopic aspect of the scaffold with 5 mm thickness. Top (**A**) and side (**B**) view, respectively; (**C**) Higher-magnification (×7.5) view of the scaffold structure; (**D**) Tomographic image of scaffold filament organization showing filaments delimiting pores; (**E**) Distribution of pores according to their size in the scaffold structure.

**Figure 2 biomedicines-11-03337-f002:**
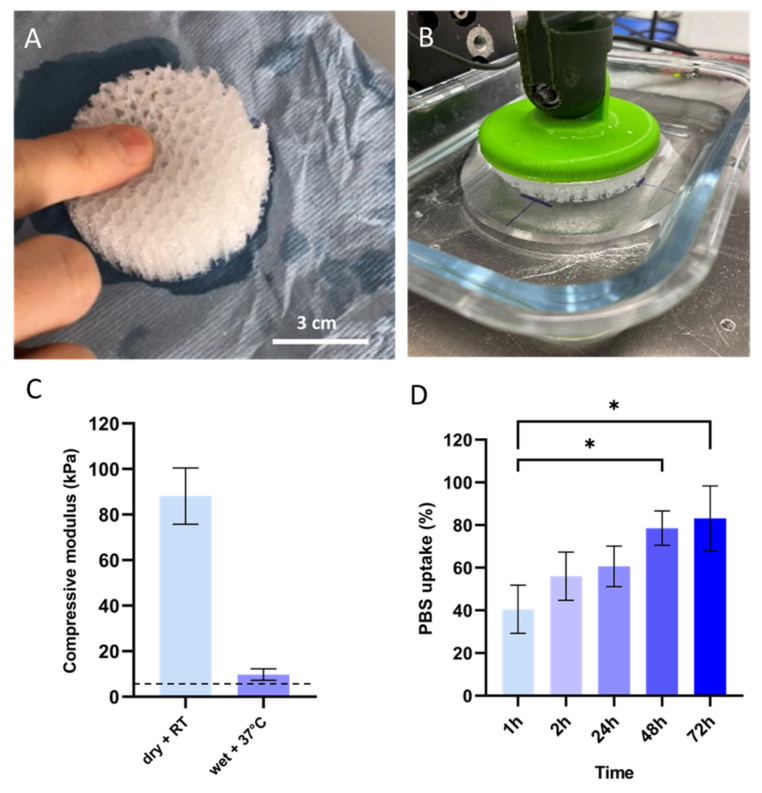
Mechanical characterization of the 3D-printed PLCL scaffold. (**A**) Visual appearance of scaffold compression after immersion in PBS at 37 °C; (**B**) Experimental set-up of the compression tray with the scaffold specimen in a 37 °C bath; (**C**) Compressive modulus of the 3D-printed PLCL scaffold in different conditions: dry at room temperature (RT) and wet at 37 °C for 5 min, data represented as mean ± SD (*n* = 3). The reference compressive modulus of human adipose tissue is indicated by a horizontal dashed line; (**D**) PBS uptake of the scaffold specimens after 1, 2, 24, 48, and 72 h in PBS at 37 °C. Data are presented as mean ± SD (*n* = 3); * *p* < 0.05.

**Figure 3 biomedicines-11-03337-f003:**
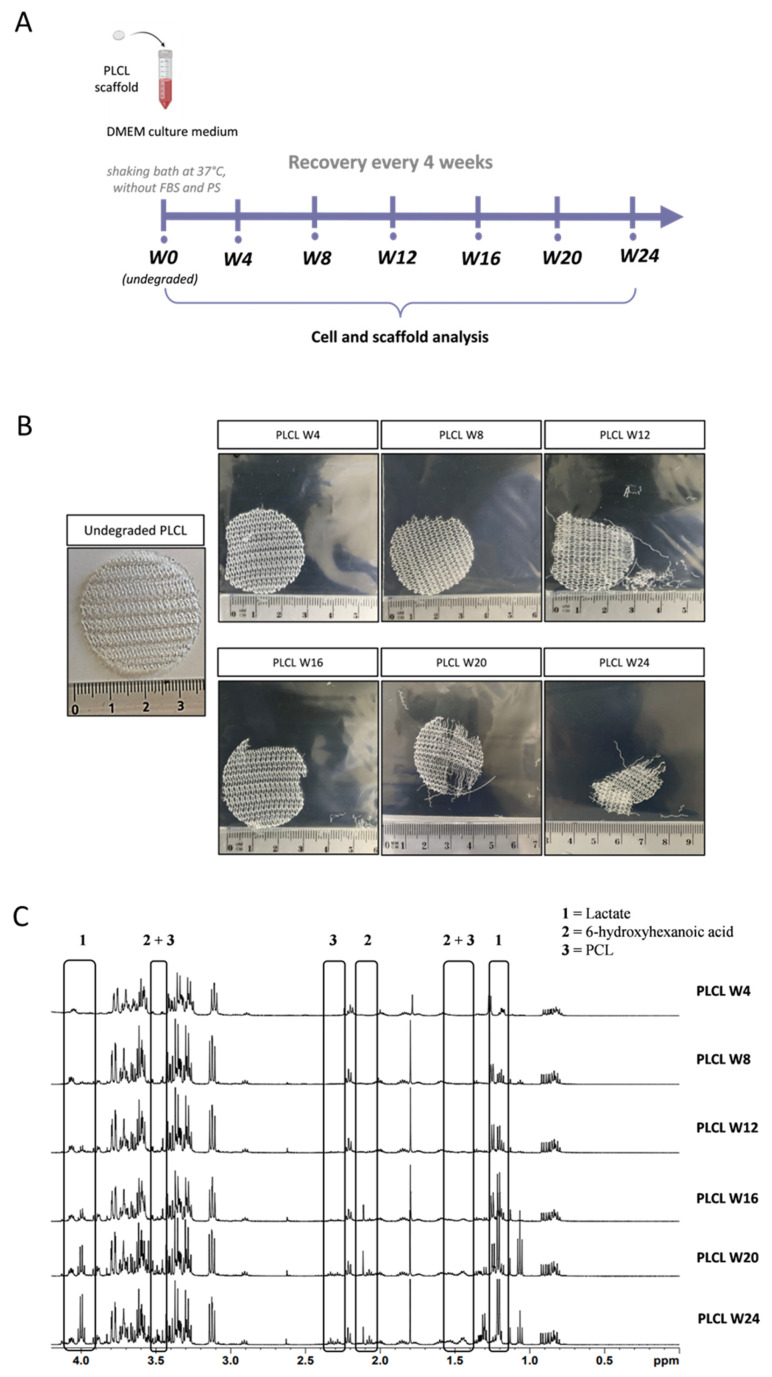
Characterization of 3D-printed PLCL scaffold degradation. (**A**) Schematic diagram depicting the in vitro degradation protocol of scaffolds in DMEM culture medium without FBS and PS (W: weeks); (**B**) Visual aspects and macroscopic changes of PLCL scaffolds before degradation (W0) and after 4 (W4), 8 (W8), 12 (W12), 16 (W16), 20 (W20), and 24 (W24) weeks of in vitro degradation; (**C**) Typical high-resolution 1D ^1^H-NMR spectra of media containing degradation products released from PLCL scaffolds at indicated times of degradation.

**Figure 4 biomedicines-11-03337-f004:**
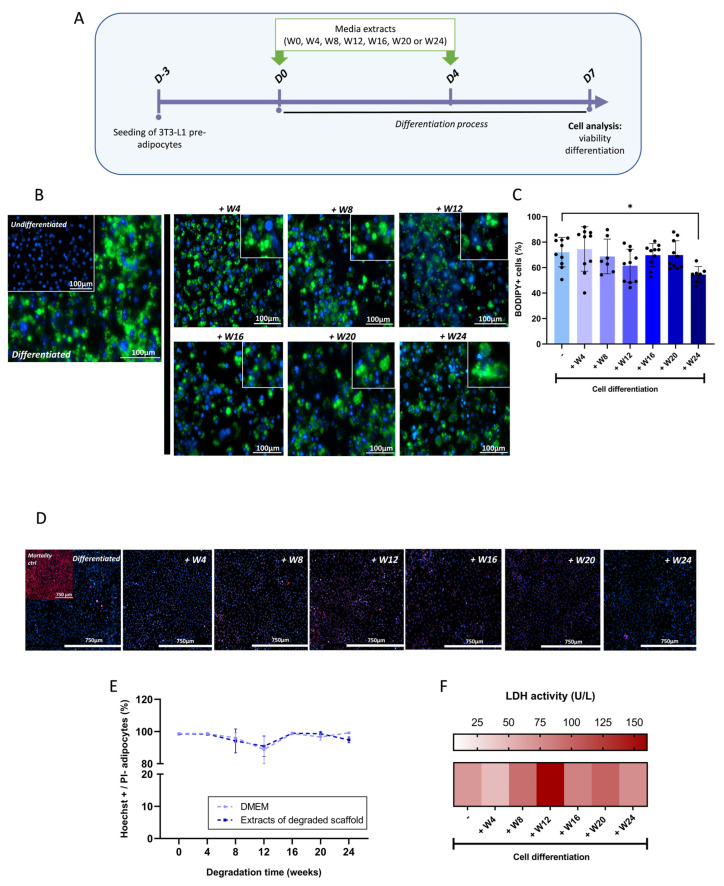
Influence of PLCL degradation products on adipocyte differentiation and viability. (**A**) Schematic diagram of the experimental protocol. Media extracts harvested at different time points of PLCL scaffold degradation (as indicated in Figure 3A) were incubated in the presence of 3T3-L1 pre-adipocytes undergoing differentiation (see Section 2) for 7 days. Then, cells were analyzed under the microscope for adipocyte differentiation (**B**) after labelling nuclear DNA with Hoechst 334342 (blue) and lipid bodies with BODIPY493/503 (green); high-magnification images were inserted at the top right corners of each image; (**C**) Percentage of BODIPY+ cells incubated for 7 days with media extracts harvested at different time points of PLCL scaffold degradation. Data were expressed as means ± SD, *n* ≥ 4, * *p* < 0.05; (**D**) Alternatively, at day 7 of differentiation in the presence of media extracts, 3T3-L1 cells were analyzed under a microscope for adipocyte viability after dual nuclear staining using Hoechst 3342 (blue fluorescent nuclei of both live dead cells) and propidium iodide (red fluorescent nuclei of dead cells). The insert at the top left corner of the first image represents the positive control (dead cells). (**E**) Evolution of the percentage of Hoechst-positive PI-negative adipocytes in the presence and absence of media extracts, as indicated in Figure 4A. Data are expressed as means ± SD, *n* = 4, *p* > 0.05. (**F**) The corresponding activity of LDH released in media in the same conditions than Figure 4E was determined. Results are expressed as heat map representation, *n* = 3.

**Figure 5 biomedicines-11-03337-f005:**
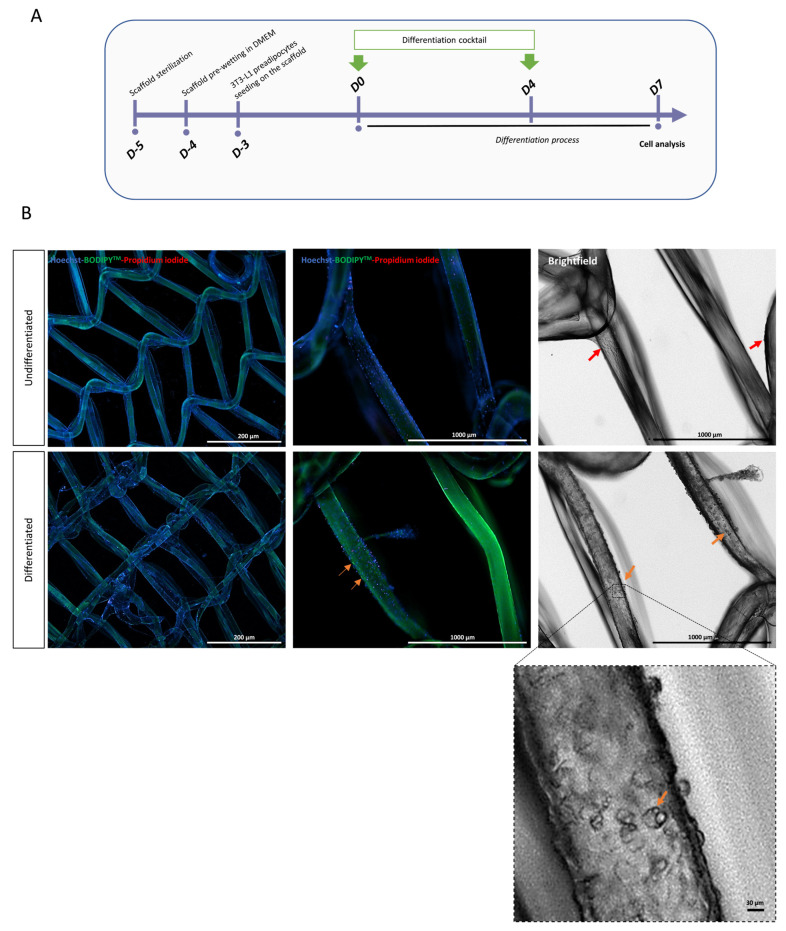
Cell attachment on 3D-printed PLCL scaffolds after 7 days. (**A**) Experimental protocol (see material and methods for details). (**B**) Representative images of undifferentiated (upper line) or differentiated (lower line) 3T3-L1 cells onto PLCL scaffolds after staining with Hoechst (blue), BODIPY 493/503 (green), and propidium iodide (red). Arrows show cells attached to scaffold after 7 days. Note the presence of lipid vesicles in the adipocytes attached to the filaments on the high-magnification image (lower panel).

**Figure 6 biomedicines-11-03337-f006:**
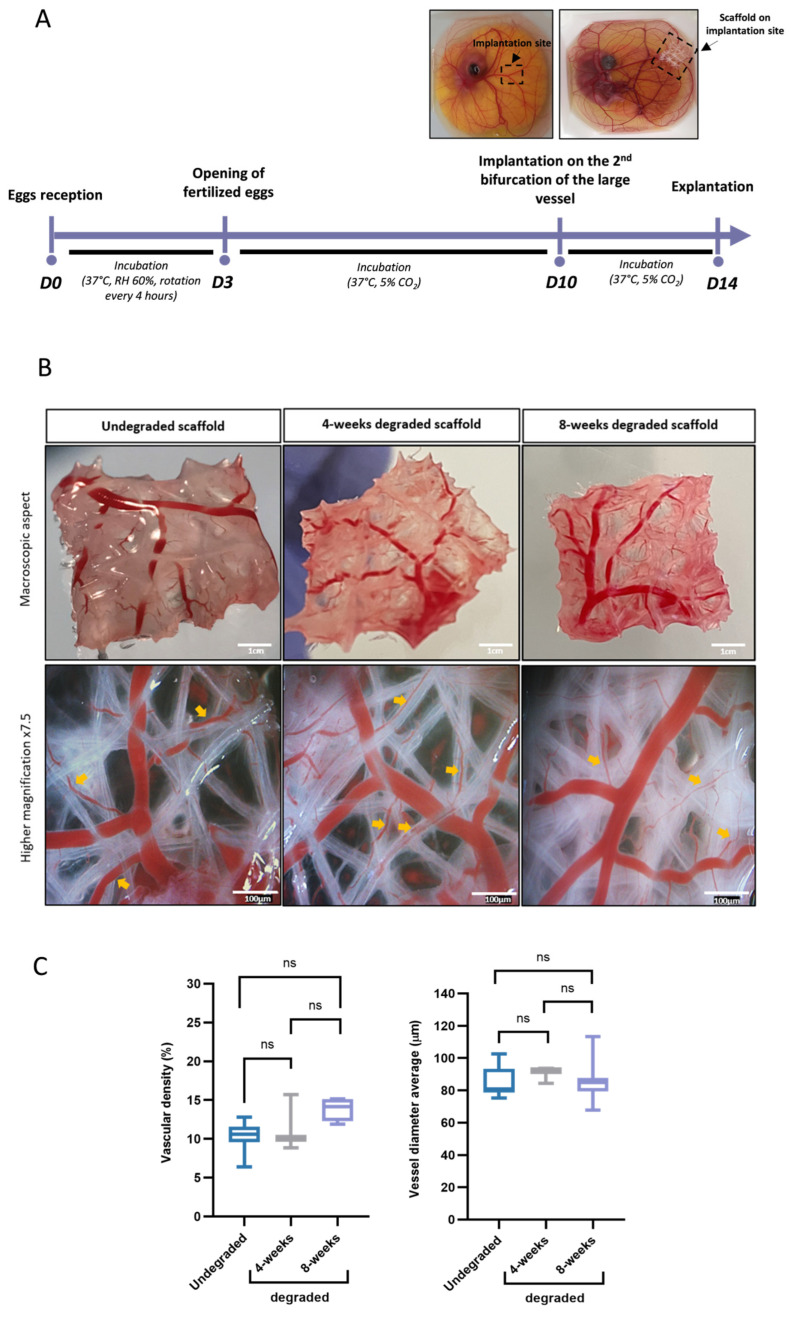
Vascularization of PLCL scaffold in the CAM assay. (**A**) Experimental diagram of the implantation of undegraded and pre-degraded scaffolds for 4 (W4) and 8 (W8) weeks on CAM (see Section 2 for details), (W: weeks); (**B**) Macroscopic (upper) and microscopic (lower) aspects of the explanted scaffolds after 4 days of implantation into CAM. Orange arrows show the orientation of the blood vessels according to the orientation of scaffold filaments; (**C**) Comparison of vascular density (left) and blood vessel diameter (right) on undegraded and pre-degraded scaffolds for 4 and 8 weeks. All data are presented as whisker bar graphs (median, 5–95 percentile max–min), *n* ≥ 4, ns *p* > 0.05.

**Figure 7 biomedicines-11-03337-f007:**
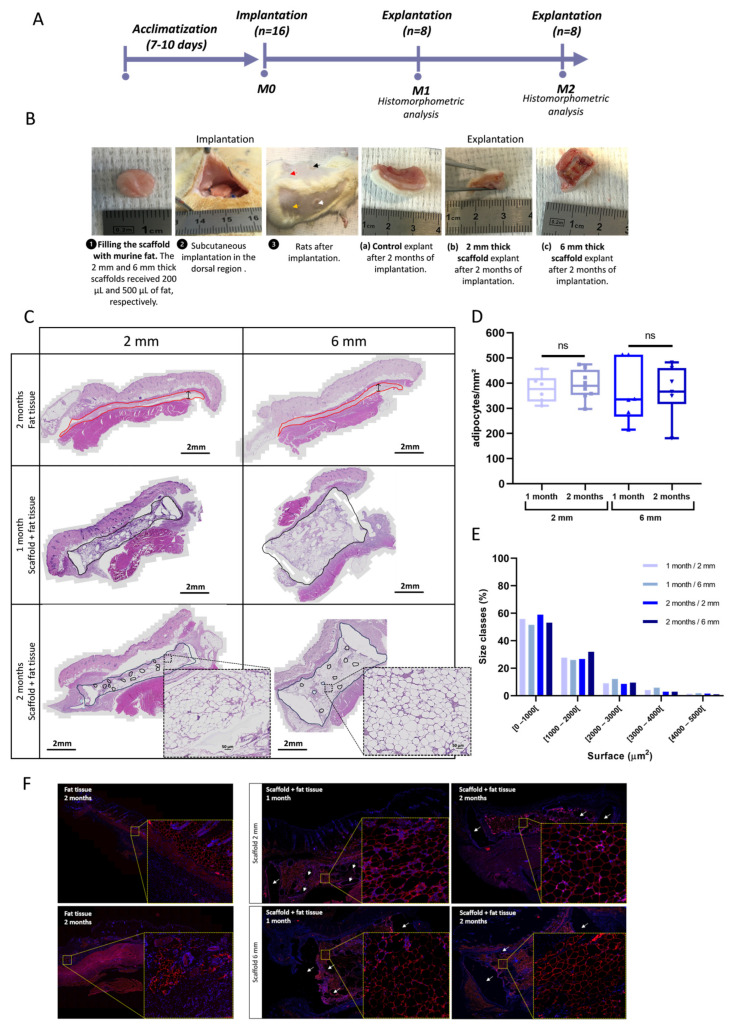
Histomorphometric analysis of adipose tissue inside scaffolds after 1 and 2 months of implantation in rats. (**A**) Experimental design. Schematic figure shows the follow-up of rats implanted with lipoaspirate-seeded PLCL scaffold (see Section 2 for details, M: months); (**B**) Surgical procedure of implanting lipoaspirate-seeded scaffolds with thicknesses of 6 mm (black arrow) or 2 mm in rats (white arrow). Controls are represented by the same volume of fat tissues implanted alone without scaffold: 0.5 mL (red arrow) and 0.2 mL (yellow arrow); (**C**) Representative image of HE staining of injected adipose tissue in the control regions and within the scaffold after 1 and 2 months of implantation. The red line surrounds the residual adipose tissue after 1 month in controls. The black line surrounds the residual adipose tissue in the scaffolds at indicated times. Higher-magnification views of adipocytes with normal morphology after 2 months of implantation are also shown; (**D**) Quantification of adipocyte number per mm^2^ inside scaffolds with thicknesses of 2 mm and 6 mm after 1 and 2 months of implantation. Data are presented as whisker bar graphs (median, 5–95 percentile max–min), *n* ≥ 6, ns *p* > 0.05; (**E**) Adipocyte size distribution (µm^2^) inside the scaffolds after 1 or 2 months of implantation; (**F**) Representative images of perilipin immunofluorescence (red staining) of adipose tissue inside the scaffolds after 1 and 2 months of implantation. Controls are represented by fat tissue implanted alone without a scaffold after 2 months of implantation. White arrows indicate the presence of scaffold filaments intertwined within adipose tissue. Higher-magnification views of a representative area are also shown.

**Figure 8 biomedicines-11-03337-f008:**
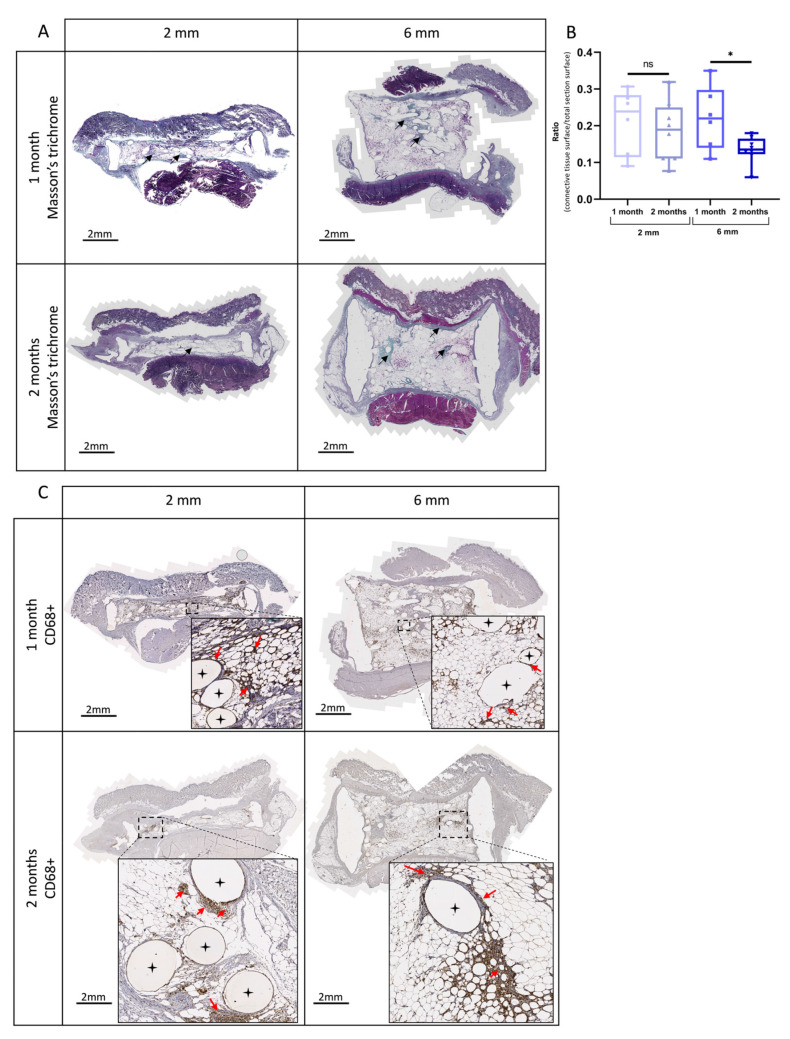
Histomorphometric analyses of connective tissue and macrophage infiltration inside the scaffolds after 1 and 2 months of implantation. (**A**) Representative images of Masson’s trichrome-stained tissue inside the scaffolds at 1 and 2 months after implantation. Dark blue areas indicate collagen fiber deposition (black arrows); (**B**) Relative quantification of connective tissue. Ratios of connective tissue to the overall surface inside scaffolds at 1 and 2 months after implantation were estimated. The whisker bar graphs indicate median, 5–95 percentile max–min, *n* ≥ 6, * *p* = 0.0320, ns *p* > 0.05; (**C**) Representative immunostaining images of infiltrating CD68^+^ macrophage inside the scaffolds (red arrows). Higher-magnification views of representative areas are also shown (brown coloration). Black stars indicate scaffold filaments.

**Figure 9 biomedicines-11-03337-f009:**
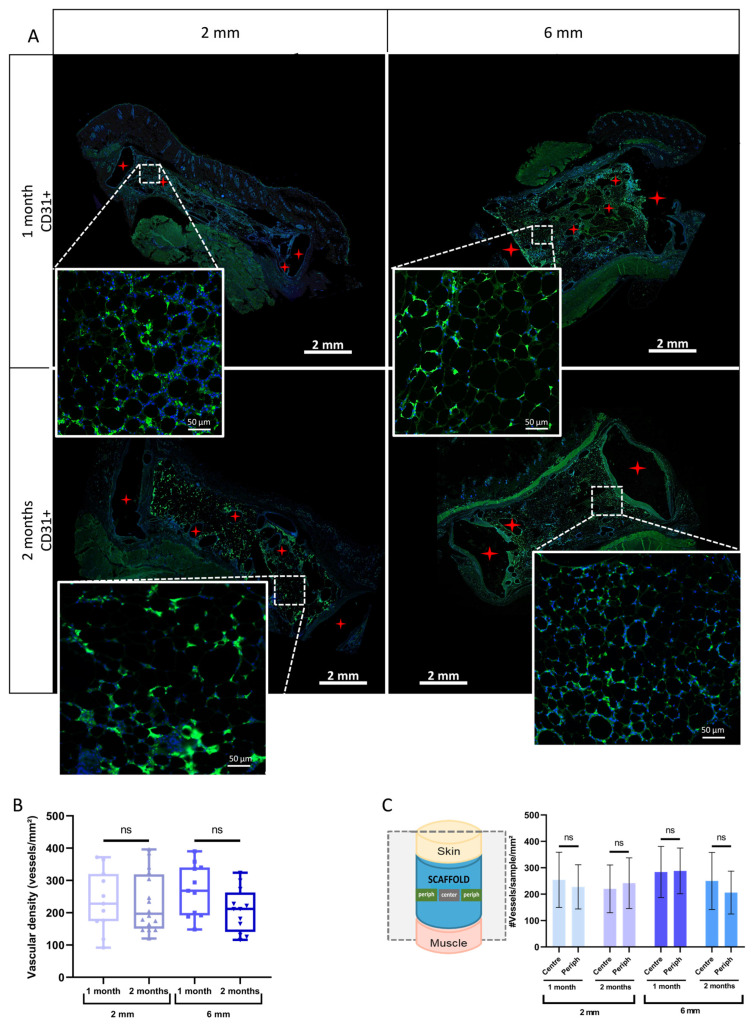
Characterization of the vascularization inside the scaffolds at 1 and 2 months after implantation. (**A**) Representative immunofluorescence images of CD31^+^ capillaries within adipose tissues inside the scaffolds. Red stars indicate scaffold pillars. Higher-magnification views of representative areas are also shown; (**B**) Quantification of CD31^+^ cells per mm^2^ (vascular density) inside the scaffolds after 1 and 2 months of implantation. Data are expressed as whisker bar graphs (median, 5–95 percentile max–min), *n* ≥ 6, ns *p* > 0.05; (**C**) Comparison of vascular density between different areas (center or on the periphery) of adipose tissue within scaffolds after 1 and 2 months of implantation. Data are presented as bar graphs (mean ± SD), *n* = 3, ns *p* > 0.05.

**Table 1 biomedicines-11-03337-t001:** Characteristics of the scaffolds used in this study.

Scaffold Characteristics		
Nature	Poly(l-lactide-co-ε-caprolactone) (PLCL)	
Bioresorbable	Yes	
3D Printer	Ultimaker S5 FDM (Utrecht, The Netherlands)	
Dimensions (length × width, cm)	25 × 10	
Nozzle diameter (mm)	0.2	
Thickness (mm)	5 (in vitro study)	
Theorical volume (mm^3^)	125.000	
Density (g/cm^3^)	<0.1	
Porosity (%)	>90%	
No. of pores/mm^2^	38	
Distribution	Heterogeneous	
**In vivo study**	**scaffold 1**	**scaffold 2**
Diameter (mm)	8	8
Thickness (mm)	2	6
Theorical volume (mm^3^)	100	300

**Table 2 biomedicines-11-03337-t002:** Resonance assignments and chemical shifts recorded from PLCL degradation products.

Molecule	Functional Group	Chemical Shift (ppm)
Lactate	CH_3_	1.21
PCL+ 6-hydroxyhexanoic acid	CH_2_	1.45–1.53
6-hydroxyhexanoic acid	CH_2_-COOH	2.07
PCL	CH_2_-COO	2.29–2.33
PCL+ 6-hydroxyhexanoic acid	CH_2_-OH	3.5
Lactate	CH-OH	4.0

## Data Availability

The data presented in this study are available on request from the corresponding author.

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
