# Peer review of "Engineering 3D-Printed Bioresorbable Scaffold to Improve Non-Vascularized Fat Grafting: A Proof-of-Concept Study"

_biomedicines, 2023, doi:10.3390/biomedicines11123337_

Round 1
Reviewer 1 Report
Comments and Suggestions for Authors
The paper provides comprehensive information on the use of 3D-printed PLCL scaffolds for adipose tissue engineering and discusses the advantages and limitations of PLCL scaffolds, including biocompatibility, degradation rate, manufacturing processes, and regulatory approval. This offers readers vital insights into the crucial factors in selecting PLCL scaffolds for adipose tissue engineering. Here are specific review comments:
1. The paper's structure is clear and well-organized. Each section provides experimental design, methods, results, and discussion to ensure that readers fully grasp various aspects of the research.
2. The language in most parts of the article is clear and precise, but there are some areas that could be further edited for readability and clarity. In particular, when describing experimental design and data analysis, ambiguity in wording should be minimized to ensure that readers completely understand the methods and results.
3. The discussion section offers a thorough analysis of the research results, highlighting the advantages and limitations of the materials used. The discussion also presents potential applications and future research directions. This is a good approach but may need to emphasize the novelty and significance of the research more to capture the reader's interest.
4. In the in vivo experiments, besides conducting safety checks on the implantation site, it is recommended to perform safety assessments on important organs of the animals to ensure the biocompatibility of the material.
5. In addition to biodegradable scaffolds, hydrogels have extensive potential applications in the field of biorepair materials. It is recommended to introduce relevant literature on hydrogels and make comparisons with biodegradable scaffold materials to make the content of the paper more comprehensive. Relevant hydrogel literature is as follows:
1) A Bionic-Homodimerization Strategy for Optimizing Modulators of Protein-Protein Interactions: From Statistical Mechanics Theory to Potential Clinical Translation
2) Biomimetic peptide dynamic hydrogel inspired by humanized defensin nanonets as the wound-healing gel coating
Reviewer 2 Report
Comments and Suggestions for Authors
The article is devoted to an interesting topic related to bioprinting of adipose tissue. Such substituents are important in the field of tissue engineering for replacing tissue defects. In general, the authors obtained a lot of data on the physicochemical and structural properties of the material, its biocompatibility and the possibility of cell adhesion.
However, there are some points that could be improved and explained.
In particular, the authors need to improve the quality of the figures. The resolution of the images should be much higher.
Introduction section should be extended by the mentioning of some recent reviews and experimental articles which were missed in this article (Biomed Mater Eng. 2020;31(4):203-210. doi: 10.3233/BME-201103. PMID: 32683340; Cyborg Bionic Syst. 2021;2021:DOI:10.34133/2021/1412542; MRS Advances 5, 855–864 (2020). https://doi.org/10.1557/adv.2020.117 and others)
It is unclear why the authors did not use cells from fat aspirate. It would be interesting to know, from the point of view of subsequent implantation, whether the scaffold will become overgrown with cells from the aspirate, and not with the own cells of experimental animals
Round 2
Reviewer 1 Report
Comments and Suggestions for Authors
I think this manuscript can be accepted.
Reviewer 2 Report
Comments and Suggestions for Authors
The authors have made all required corrections. The article may be accepted for publication